

# The role of $^{18}$F-FDG PET/CT in patients with serous cavity effusion of undetermined origin: a retrospective clinical study

Xianwen Hu, Ya Li, Jiong Cai and Pan Wang

Department of Nuclear Medicine, Affiliated Hospital of Zunyi Medical University, Zunyi, Guizhou, China

## ABSTRACT

**Aim**. The diagnostic performance of positron emission tomography with fluoro-18 fluorodeoxyglucose integrated with computed tomography ($^{18}$F-FDG PET/CT) in identifying the primary cause of unknown serous effusion and malignant tumors with serous metastasis was evaluated in our study.

**Methods**. A retrospective analysis was conducted on 134 patients with unexplained serous cavity effusion, including pericardial effusion, pleural effusion, and ascites, who underwent $^{18}$F-FDG PET/CT scans. The cohort comprised 94 cases of malignant disease and 40 cases of benign disease. Visual analysis of all $^{18}$F-FDG PET/CT images and semi-quantitative analysis by measuring maximum standardized uptake value (SUVmax) in the region of interest were performed. The diagnostic capabilities of SUVmax, Ca125, Ca199, and serum carcinoembryonic antigen were compared by plotting the areas under the receiver operating characteristic curve.

**Results**. The primary disease of serous cavity effusion was diagnosed with a sensitivity of 90.1%, specificity of 78.8%, and accuracy of 85.7% using $^{18}$F-FDG PET/CT. The SUVmax of primary malignant lesions was found to be significantly higher than that of benign lesions, with values of $12.83 \pm 6.64$ and $4.48 \pm 3.16$ ($P < 0.001$), respectively. The detection of serous cavity metastasis by PET/CT showed a sensitivity of 84.3%, specificity of 94.0%, and accuracy of 88.3%. In receiver operating characteristic (ROC) analysis, the area under the curve of SUVmax was the largest ($P < 0.01$), significantly surpassing that of serum Ca125, Ca199, and CEA.

**Conclusion**. $^{18}$F-FDG PET/CT was determined to be an effective imaging modality for detecting undetermined serous cavity effusion, exhibiting high diagnostic performance in the differentiation of primary disease from benign to malignant and the evaluation of serous cavity metastasis.

Corresponding authors
Jiong Cai, jiong_cai@163.com
Pan Wang, 1298178828@qq.com

## INTRODUCTION

The serous cavity encompasses the chest, abdominal, and pericardial cavities. Under normal circumstances, a minimal quantity of fluid resides in the serous cavity, serving to lubricate and shield organs and tissues. Pathological conditions such as inflammation, trauma, tumors, autoimmune, and metabolic diseases can lead to an accumulation of fluid in the

serous cavity, termed serous cavity effusion (SCE) (*Rossi et al., 2015*; *Gecmen et al., 2018*). Based on the underlying cause, SCE can be benign or malignant. Common benign causes include cardiac insufficiency, decompensation of liver cirrhosis, non-specific inflammation, renal failure, tuberculosis, and similar conditions (*Lee et al., 2014*). Malignant serous cavity effusion arises in the advanced stages of various malignancies due to tumor invasion of blood vessels, lymphatic vessels, or lymph node metastasis, leading to the blockage of serous effusion absorption or direct tumor cell invasion of blood vessels, enhancing capillary permeability (*Porcel, 2018*). While puncturing the serous cavity for effusion extraction and cytological examination is the gold standard for determining the nature of SCE, its sensitivity is only 30%–50%, and it poses certain risks as an invasive procedure. Traditional imaging methods, including ultrasound, computed tomography (CT), and magnetic resonance imaging (MRI), can identify the presence and estimate the amount of serosal effusion. They aid in differentiating between benign and malignant diseases of the chest and abdominal cavity by examining signs of benign and malignant effusion. However, ultrasonography, CT, and MRI primarily focus on local anatomical structures, making it challenging to identify small lesions and hidden metastases, limiting their ability to differentiate and diagnose the nature of serosal effusion (*Sugiyama et al., 2011*). Positron emission tomography (PET)/CT imaging, combining anatomical CT with functional PET, has played a pivotal role in diagnosing, staging, and assessing tumor prognosis post-treatment. It is increasingly valued for identifying and diagnosing the nature of serosal effusion (*Zhang et al., 2009*; *Li et al., 2013*; *Porcel, 2018*). This study aims to investigate the utility of positron emission tomography with fluoro-18 fluorodeoxyglucose integrated with computed tomography ($^{18}$F-FDG PET/CT) in localizing and qualitatively diagnosing primary lesions in unexplained serous cavity effusion, as well as assessing the diagnostic performance for serous cavity metastasis in malignant tumors.

## MATERIAL AND METHODS

This retrospective study received approval from the Ethics Committee of Affiliated Hospital of Zunyi Medical University (approval number: KLL-2022-166), and all methods were performed in accordance with the relevant guidelines and regulations.

### Patients

Medical records of patients who underwent $^{18}$F-FDG PET/CT examinations from December 2019 to June 2023 were retrospectively reviewed to identify the primary cause of serous cavity effusion, including pericardial effusion, pleural effusion, and ascites.

Enrollment criteria: (i) The cause of serous cavity effusion in enrolled patients remained uncertain after preliminary examinations such as CT, ultrasound, blood tests, or cytological analysis. The nature (benign or malignant) of the serous cavity effusion was also undetermined. (ii) Alternatively, patients with negative cytological results and abnormal lesions identified through non-invasive examinations, where the main cause of serous cavity effusion couldn't be determined. (iii) Patients had no history of prior tumor-related surgeries or other treatments before examination. (iv) All enrolled patients

underwent serous cavity effusion cytology, surgical pathology, or CT, MRI, PET/CT imaging follow-up, ultimately confirming the diagnosis.

### $^{18}$F-FDG PET/CT imaging

The method of production of $^{18}$F-FDG and PET/CT imaging was carried out in accordance with our previously published literature (*Hu et al., 2024*).

### Imaging analysis

All PET/CT images were interpreted by two attending nuclear medicine physicians, each possessing over 5 years of experience in PET/CT imaging diagnosis. In the event of discrepancies or disagreements, both physicians engaged in negotiations until a consensus was achieved. Any abnormal focal lesion exhibiting an increase in $^{18}$F-FDG uptake greater than the background activity of the corresponding organ tissue (excluding physiological uptake in the gastrointestinal or visceral regions) was considered a potential positive lesion site. Semi-quantitative analysis involved calculating the maximum standardized uptake value (SUVmax) for the region of interest (ROI) in patients with benign and malignant serous cavity effusion (SCE). In cases where a focal lesion with increased $^{18}$F-FDG uptake was present in the serous cavity, the SUVmax of the lesion was directly measured. If there was no focal uptake in the serous cavity effusion, a circular ROI with a diameter of five mm from the serosal wall was placed, and the SUVmax of the serous cavity effusion was recorded. A true positive localization diagnosis of the primary tumor on PET/CT imaging was defined if an abnormal lesion identified on PET/CT (or CT alone) was confirmed by pathology as a malignant/benign tumor with serous cavity effusion. Conversely, if abnormal lesions detected by PET/CT (or CT alone) were not confirmed by pathology, the result was considered false positive. A true negative PET/CT localization diagnosis was defined when no suspicious lesions were found on PET/CT (or individual CT), or if the lesions were non-neoplastic and confirmed by pathology or follow-up. If the PET/CT (or individual CT) result was negative, but a tumor was later discovered, the PET/CT result was deemed false negative.

### Statistical analysis

The statistical analysis was performed using SPSS version 18.0 (IBM Corporation, Armonk, NY, USA). Normality of continuous variables was assessed using the Shapiro–Wilk test. If the data exhibited a normal distribution, they were expressed as mean ± standard deviation, and multiple group comparisons were conducted using one-way ANOVA. For non-normally distributed data, the statistical description was presented as the median (Q1, Q3), and between-group comparisons were performed using the independent samples rank sum test. Categorical data were presented as the number of cases (%) for count variables, and between-group comparisons were carried out using the chi-square test. In cases where the conditions for the chi-square test were not met, Fisher's exact probability method was applied. Correlation analysis was conducted using Spearman's method, and the results were visualized through scatter plots. All tests were two-sided, and statistical significance was considered when $p < 0.05$.

## RESULTS

### Distribution of primary diseases

A total of 134 patients, including 60 men and 74 women, with unexplained serous cavity effusion underwent PET/CT evaluation, with a mean age of 58.8 ± 13.4. Among these patients, there are seven with pericardial effusion (7/134), 61 with pleural effusion (61/134), 35 with ascites (35/134), 13 with both pericardial effusion and pleural effusion (13/134), 17 with both pleural effusion and ascites (17/134), and 1 patient with pericardial effusion, pleural effusion, and ascites (1/134). Primary diseases included 101 tumor lesions comprising 94 malignant and seven benign tumors, and 33 non-tumor lesions. Details of the distribution of primary diseases are presented in Table 1.

### Detection of primary lesions

Among the 134 patients with unexplained serous cavity effusion, 98 cases were abnormal. Of the 98 abnormal scans, 91 were eventually diagnosed with malignant lesions. For the remaining 36 scans, 26 benign lesions were accurately identified, and 10 malignant lesions were missed by PET/CT. The positive predictive rate (PPV) was 92.9%, and the negative predictive rate (NPV) was 72.2%. Typical true positive and true negative cases are demonstrated in (Figs. S1 and S2). False-negative results were pathologically confirmed in 10 cases, including two cases of unknown origin (as shown in Fig. 1), one case of pericardial mesothelioma, one case of pleural mesothelioma, one case of lung cancer, two cases of advanced adenocarcinoma of the fallopian tubes, one case of liver cancer, one case of gastric mucosa-associated lymphoid tissue (MALT) lymphoma, and one case of gallbladder signet ring cell carcinoma. Seven cases were confirmed falsely positive after surgery or biopsy, including two cases of ovarian theca fibroma showing focal increased $^{18}$F-FDG uptake on PET/CT, misdiagnosed as ovarian cancer, two cases of organized pneumonia showing strongly increased $^{18}$F-FDG uptake, misdiagnosed as lung cancer, one case of tuberculous pericardium misdiagnosed as malignant mesothelioma (as shown in Fig. 2), one case of tuberculous peritonitis misdiagnosed as ovarian cancer with peritoneal metastasis (as shown in Fig. 3), and one case of Castleman's disease, misdiagnosed as lymphoma. For the primary cause of pericardial effusion, PET/CT demonstrated sensitivity, specificity, and accuracy of 87.5%, 84.6%, and 85.7%, respectively. In cases of pleural effusion, the sensitivity, specificity, and accuracy were 91.4%, 86.4%, and 90.2%, respectively. For ascites, PET/CT exhibited sensitivity, specificity, and accuracy of 88.4%, 80.0%, and 86.8%, respectively. Overall, the sensitivity, specificity, and accuracy of PET/CT in diagnosing the etiology of serous cavity effusion were 90.1%, 78.8%, and 85.7%, respectively (refer to Table 1). The SUVmax of primary malignant lesions was significantly higher than that of benign lesions (12.83 ± 6.64 *vs.* 4.48 ± 3.16, $P < 0.001$) (see Fig. 4).

### Detection of pericardium, pleura or/and peritoneum metastasis

Among the 94 cases of primary malignant lesions, 51 cases were accompanied by pericardial, pleural, or peritoneal metastasis, and 43 of these cases were detected through PET/CT (43/51). The sensitivity, specificity, and accuracy of PET/CT in the detection of serous cavity metastasis were 84.3%, 94.0%, and 87.3%, respectively (refer to Table 2).

**Table 1  Clinical and epidemiologic features of patients with serous cavity effusion of undetermined origin.**

| Parameters | Pericardial effusion (N = 21)[a] | Pleural effusion (N = 92)[b] | Ascites (N = 53)[c] | Total (N = 134) |
|---|---|---|---|---|
| Gender | | | | |
| Man | 12 (57.1%) | 51 (55.4%) | 11 (20.8%) | 60 (44.8%) |
| Woman | 9 (42.9%) | 41 (44.6%) | 42 (79.2%) | 74 (55.2%) |
| Age (years) | $61.9 \pm 13.6$ | $59.4 \pm 14.5$ | $56.5 \pm 12.7$ | $58.8 \pm 13.4$ |
| Primary lesions | | | | |
| Tumor lesions | 8 (38.1%) | 70 (70.6%) | 43 (81.1%) | 101 (75.4%) |
| Malignant tumors | 6 (28.6%); including mesothelioma (N = 2), lung cancer (N = 2), liver cancer (N = 1), lymphoma (N = 1) | 64 (65.2%); including lung cancer (N = 38), lymphoma (N = 9), ovarian cancer (N = 9), mesothelioma (N = 2), thymic carcinoma (N = 2), prostate cancer (N = 1), rectal cancer (N = 1), unknown origin (N = 2). | 40 (75.5%), including ovarian cancer (N = 25), lymphoma (N = 5), adenocarcinoma of unknown origin (N = 2), intestinal cancer (N = 2), pancreatic cancer (N = 1), gallbladder carcinoma (N = 2), mesothelioma (N = 1), liver caner (N = 2) | 94 (70.1%) |
| Benign tumors | 2 (9.5%); including thymoma type A (N = 1) and theca fibroma (N = 1) | 6 (5.4%); including ovarian sex gonad stromal tumor (N = 4), thymoma type A (N = 2). | 3 (5.7%); ovarian sex gonad stromal tumor (N = 3) | 7 (5.2) |
| Non-tumorous lesions | 13 (61.9%) | 22 (23.9%) | 10 (18.9%) | 33 (24.6%) |
| Nonspecific inflammation[d] | 6 (28.6%) | 11 (12.0%) | 1 (1.9%) | 13 (9.7%) |
| Tuberculosis | 2 (9.5%) | 5 (5.4%) | 4 (7.5%) | 5 (3.7) |
| Others | 5 (23.8%); including dilated cardiomyopathy (N = 2); hypothyroidism (N = 1); heart failure (N = 2) | 6 (6.5%); including hypothyroidism (N = 1), heart failure (N = 3), liver cirrhosis (N = 2). | 5 (9.4%); including liver cirrhosis (N = 4), Castleman's disease (N = 1). | 15 (11.2%) |

**Notes.**

[a] Including 13 cases of pericardial effusion combined with pleural effusion and 1 case of pericardial effusion combined with pleural effusion and ascites.

[b] Including 13 cases of pleural effusion combined with pericardial effusion, 17 cases of pleural effusion combined with ascites and 1 case of pericardial effusion combined with pleural effusion and ascites.

[c] Including 17 cases of ascites combined with pleural effusion, and 1 case of ascites combined with pericardial effusion and pleural effusion.

[d] Including three patients with pericardial effusion, five with pleural effusion, and one with both pericardial and pleural effusion, all underwent re-examination after anti-inflammatory treatment. The subsequent follow-up showed that the lesions had disappeared, and no recurrence was observed for more than half a year. T and PET/CT in Detection of Primary Lesion.

Of the 43 patients, 41 were true positive for both primary and metastases on PET/CT, and only two were true positive for pleural and peritoneal metastases but false negative for primary lesions. Among the eight false-negative patients, four had false negatives for primary tumors and metastatic lesions on PET/CT, while the other four had positive primary lesions but false-negative metastatic lesions, including two lung cancers, one ovarian cancer, and one rectal adenocarcinoma. Three false-positive patients included two cases of lung cancer with bacterial pleurisy and one case of liver cancer with bacterial peritonitis.

The SUVmax of malignant serous cavity metastasis was significantly higher than that of benign serous cavity effusion ($8.88 \pm 4.19$ *vs.* $2.09 \pm 1.50$, $P < 0.001$) and primary benign lesions ($8.88 \pm 4.19$ *vs.* $4.48 \pm 3.16$, $P < 0.01$). Figure 4 illustrates that in benign lesions, except for some non-specific inflammations and tuberculosis where the SUVmax can reach

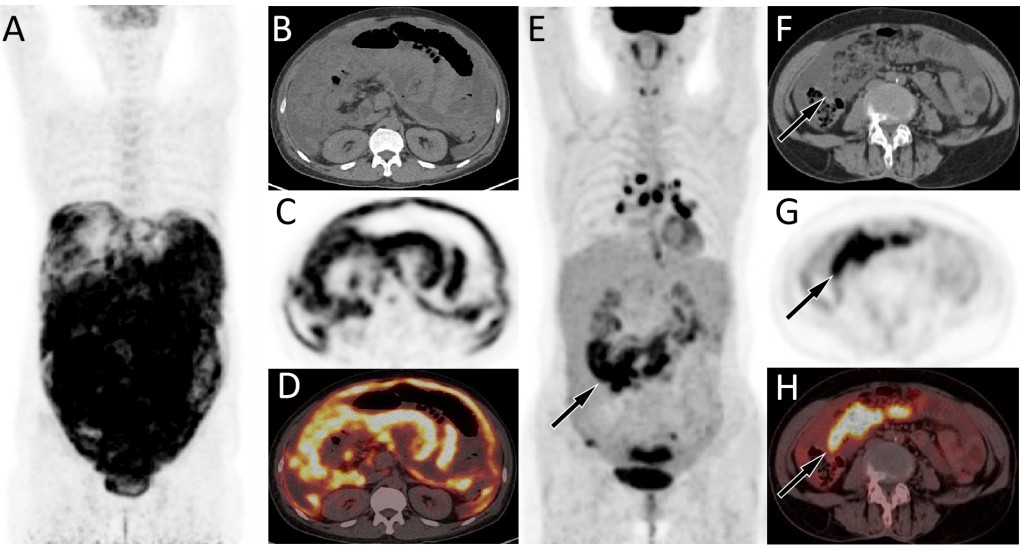

**Figure 1** **(A–D) PET/CT images of 44-y-old man who presented with ascites and pleural effusion. (E–H) PET/CT images of 67-y-old woman who presented with ascites and pleural effusion.** The patient's serum Ca125 was 1345.0, with normal carcinoembryonic antigen and Ca199 levels, and history of abdominal pain and bloating for more than 1 month. The MIP image (A) showed a diffuse increase in $^{18}$F-FDG uptake in the abdomen. Axial CT (B) showed uneven thickening of the visceral and parietal peritoneum; axial PET (C) and axial fused PET/CT (D) showed an increase in $^{18}$F-FDG uptake at the corresponding thickened peritoneum, with a SUVmax of 7.4 (arrows), suggesting manignant lesions. Subsequently, adenocarcinoma metastasis was confirmed by pathology, but the primary focus has not been found. (E–H) PET/CT images of 67-y-old woman who presented with ascites and pleural effusion. The patient's serum Ca125 was 1691.0, with normal carcinoembryonic antigen and Ca199 levels, and history of chest and abdominal pain for over 1 month. The MIP image (E) showed the right abdominal region presented an uneven increase in $^{18}$F-FDG uptake. Axial CT (F) showed fluid accumulation in the abdominal cavity with right visceral peritoneal thickening (arrow); axial PET (G) and axial fused PET/CT (H) showed an increase in $^{18}$F-FDG uptake at the corresponding site, with a SUVmax of 15.6 (arrows), suggesting manignant lessions. Subsequently, cytological examination of ascites revealed malignant tumor cells, but exploratory laparotomy did not reveal the origin of the primary lesion.

around 10.0, most benign lesions or benign serous effusions have lower SUVmax values, often below 3.0. In contrast, in most malignant primary lesions or metastatic lesions, the SUVmax is usually above 5.0.

## Receiver operating characteristic curves

The receiver operating characteristic (ROC) curves of different modalities for differentiating malignant from benign serous cavity effusion are presented in Fig. 5. The areas under the curve for SUVmax, Ca125, Ca199, and serum CEA were 0.845 ($P < 0.001$), 0.631 ($P = 0.073$), 0.524 ($P = 0.728$), and 0.518 ($P = 0.787$), respectively. The highest accuracy was calculated to be 87.3% when the SUVmax of malignant serous cavity effusion was greater than 1.8, and the area under the curve (AUC) for this was the largest ($P < 0.01$), significantly exceeding that of serum Ca125, Ca199, and CEA.

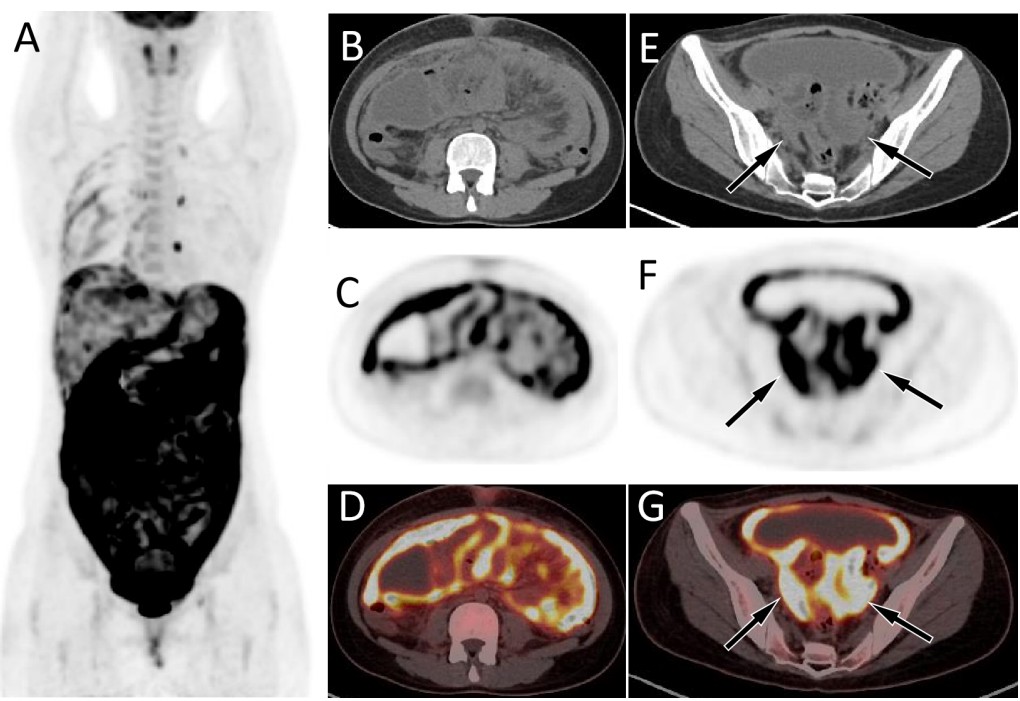

**Figure 2** **PET/CT images of 26-y-old woman who presented with ascites.** The patient's serum Ca125 was 703.0, with normal carcinoembryonic antigen and Ca199 levels, and history of abdominal pain, bloating, poor appetite for 1 month. The MIP image (A) showed a diffuse increase in $^{18}$F-FDG uptake in the abdomen. Axial CT (B) showed uneven thickening of the visceral and parietal peritoneum; axial PET (C) and axial fused PET/CT (D) showed an increase in $^{18}$F-FDG uptake at the corresponding thickened peritoneum, with a SUVmax of 12.4 (arrows). Moreover, soft tissue mass shadows can be seen in the bilateral adnexal areas on axial CT (E, arrows); There are also strongly increased $^{18}$F-FDG uptake on axial PET (F) and PET/CT (G). These PET/CT findings mimic the appearance of ovarian cancer with diffuse peritoneal implantation metastasis. However, the pathological results after puncture biopsy showed that was tuberculous peritonitis.

## DISCUSSION

Both benign and malignant conditions can lead to serous cavity effusion. In this study, malignant tumors causing pericardial effusion included mesothelioma, lymphoma, while benign conditions comprised heart failure, tuberculous pericarditis, and type A thymoma. Malignant tumors associated with pleural effusion included lung cancer, ovarian cancer, and lymphoma, with benign conditions primarily featuring tuberculous pleurisy and non-specific inflammation. The leading causes of ascites involved ovarian cancer, gastrointestinal and biliary cancers, whereas benign conditions encompassed tuberculous peritonitis, non-specific inflammation, and ovarian sex cord stromal tumors.

Routine examination methods for serous cavity effusion include biochemical analysis, cytological examination, and traditional imaging techniques such as X-ray, B-ultrasound, and CT. However, the positive rate of cytological examination for serous cavity effusion is low, and the sensitivity and specificity of tumor markers in blood and serous cavity effusion are limited (*Naito, Shimada & Yuki, 2022*). X-ray, ultrasound, and CT have a

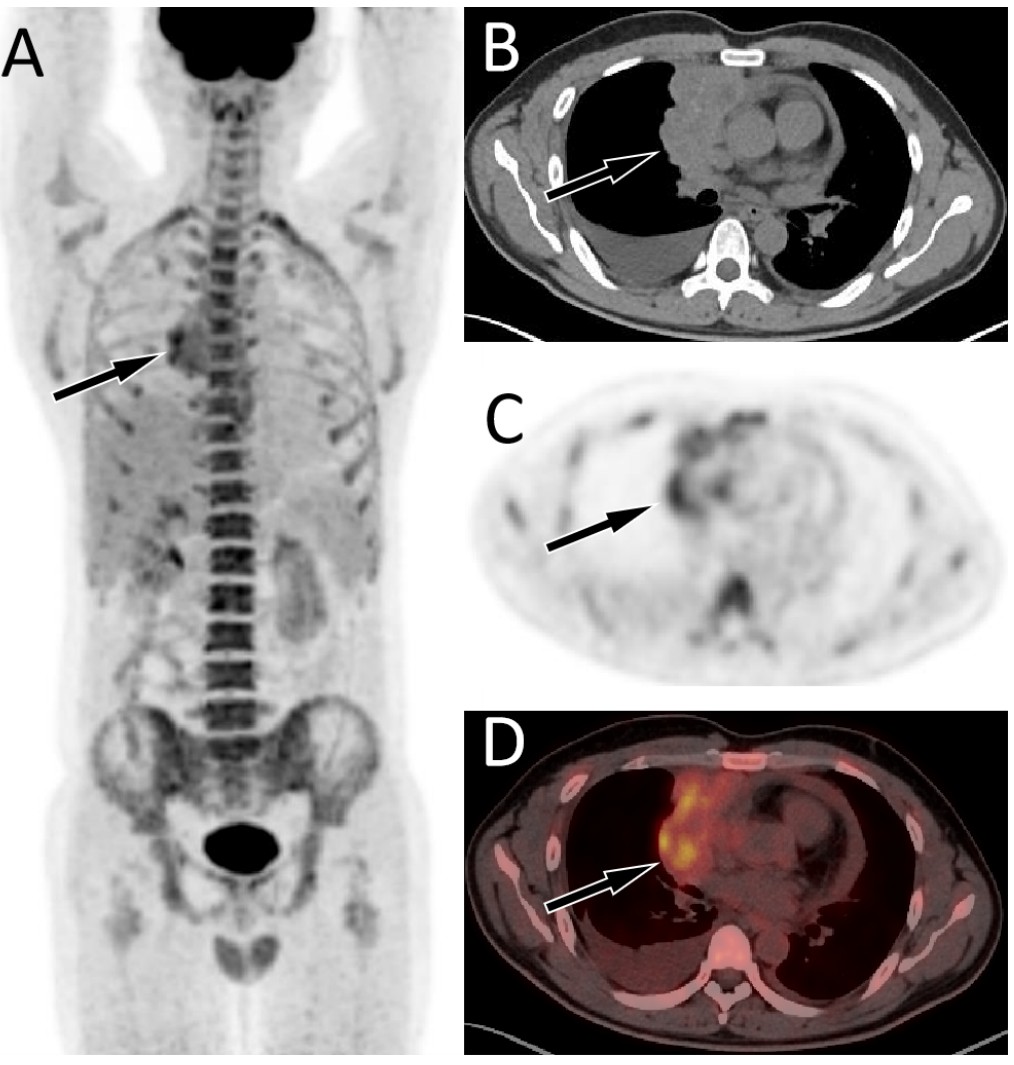

**Figure 3** **PET/CT images of 34-y-old man who presented with pericardial effusion and pleural effusion.** The patient's serum Ca125 was 181.0, with normal carcinoembryonic antigen and Ca199 levels, and history of chest tightness and breathing difficulties 2 weeks ago. The maximum intensity projection (MIP, A) revealed an increased $^{18}$F-FDG uptake in the right margin of the heart. Axial CT (B) showed an uneven soft tissue density mass on the right side of the pericardium; axial PET (C) and axial fused PET/CT (D) showed an increase in $^{18}$F-FDG uptake at the corresponding site, with a SUVmax of 5.7 (arrows), mimicking a malignant lesion. However, the pathological results after puncture biopsy showed that the lesion was tuberculosis.

restricted role in the differential diagnosis of diseases affecting the pericardium, pleura, and peritoneum. Some patients exhibit atypical medical histories and clinical manifestations, making it challenging to determine the etiology of serous effusion through various routine examinations (*Strange et al., 2023*). Malignant tumor cells demonstrate an increased rate of glycolysis, a characteristic reflected in strong $^{18}$F-FDG uptake on PET/CT. Serous cavity effusion is a multisystem disorder resulting from various conditions throughout the body. PET/CT can provide whole-body imaging, allowing for the simultaneous localization of

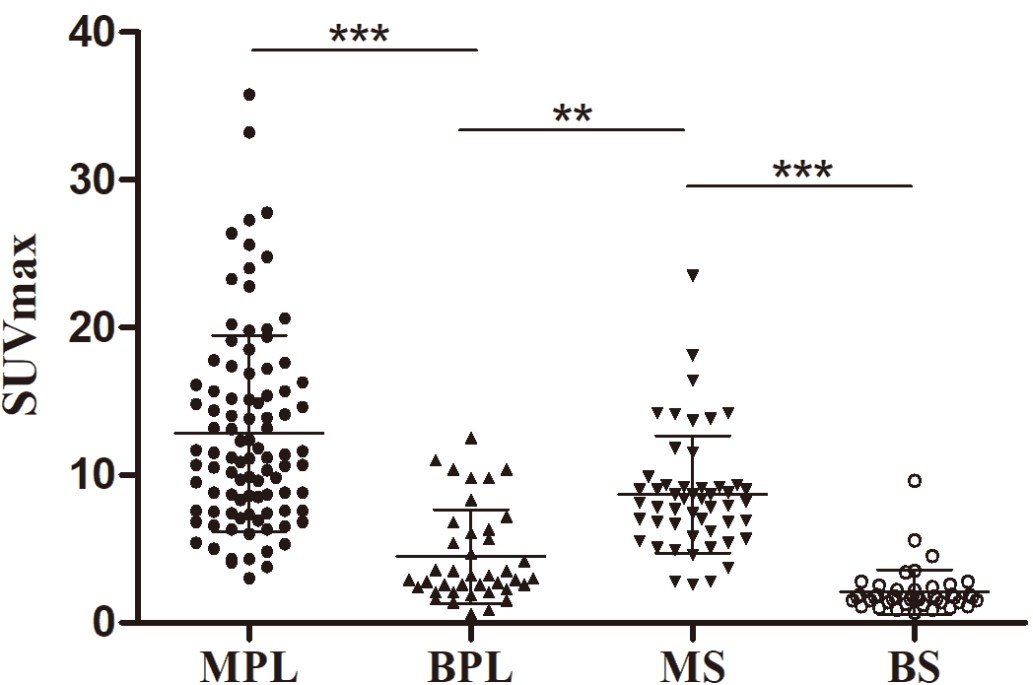

**Figure 4 Comparison of SUVmax in manignant primary lessions (MPL), benign primary lessions (BPL), malignant serous cavity effusion (MS) and benign serous cavity effusion (BS).** $**p < 0.01$; $***p < 0.001$.

**Table 2 Results of PET/CT in detection of primary lesions causing serous cavity effusion.**

| Index | TP | FN | TN | FP | Sensitivity (%) | Specificity (%) | PPV (%) | NPV (%) | Accuracy (%) |
|---|---|---|---|---|---|---|---|---|---|
| Pericardial effusion ($N = 21$) | 7 | 1 | 11 | 2 | 87.5 | 84.6 | 77.8 | 91.7 | 85.7 |
| Pleural effusion ($N = 92$) | 64 | 6 | 19 | 3 | 91.4 | 86.4 | 95.5 | 76.0 | 90.2 |
| Ascites ($N = 53$) | 38 | 5 | 8 | 2 | 88.4 | 80.0 | 95.0 | 61.5 | 86.8 |
| Total ($N = 134$) | 91 | 10 | 26 | 7 | 90.1 | 78.8 | 92.9 | 72.2 | 87.3 |
| Serous cavity metastasis | 43 | 8 | 40 | 3 | 84.3 | 93.0 | 93.5 | 83.3 | 88.3 |

Notes.
TP, true-positive; FN, false-negative; TN, true-negative; FP, false-positive; NPV, negative predictive value; PPV, positive predictive value.

primary and metastatic lesions, showcasing high sensitivity in various malignant tumors (*Zhang et al., 2009*; *Chen et al., 2016*). Consequently, PET/CT serves as a suitable imaging modality for diagnosing unexplained serous cavity effusion.

For clinical doctors and patients, early diagnosis of the primary cause of serous cavity effusion is crucial for formulating correct clinical treatment plans and predicting prognosis. While a few previous studies have discussed the role of PET/CT in detecting the primary causes of pleural and ascites, revealing good diagnostic performance (*Chen et al., 2016*; *Yang et al., 2019*; *Lu et al., 2022*; *Simó et al., 2023*), our study is the first to systematically evaluate the role of PET/CT in the diagnosis of unexplained serous cavity effusion and further assess its diagnostic performance in pericardial effusion, pleural effusion, and peritoneal effusion. Overall, our study revealed that PET/CT in locating the primary

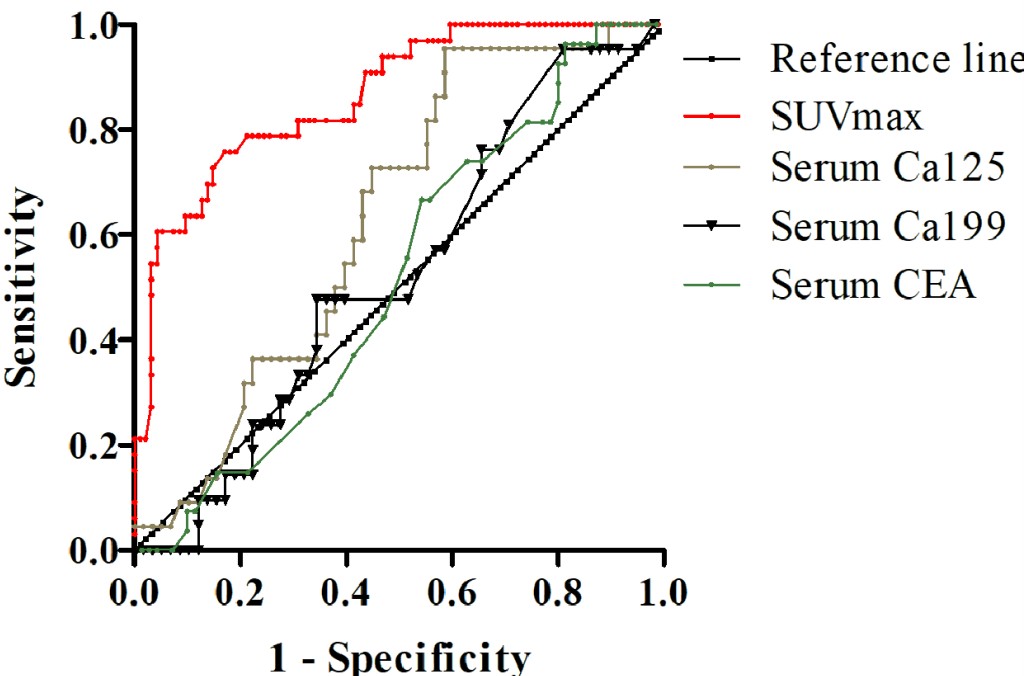

**Figure 5  ROC curve of SUVmax, serum carbohydrate antigen 125 (Ca125), Ca199, and carcinoembryonic antigen (CEA) in differential diagnosis of serous cavity effusion.**

disease site of serous cavity effusion had high diagnostic performance, with a sensitivity of 90.1%, specificity of 78.8%, and accuracy of 87.3%. Further analysis based on the location of the effusion showed that the sensitivity, specificity, and accuracy of PET/CT in determining the primary cause of pericardial effusion were 87.5%, 84.6%, and 85.7%, respectively. It is noteworthy that PET/CT correctly identified a patient with pericardial effusion induced by ovarian thecoma, who was also accompanied by pleural and ascitic fluid, known as Meigs syndrome. After undergoing surgery to remove the tumor, all serous cavity effusions, including pericardial effusion, disappeared, confirming our diagnosis. For determining the primary cause of pleural effusion, the sensitivity, specificity, and accuracy of PET/CT were 91.4%, 86.4%, and 90.2%, respectively, similar to the results of *Yang et al. (2019)*. Regarding ascites, PET/CT demonstrated a sensitivity of 88.4%, specificity of 80.0%, and accuracy of 86.8% in determining the primary cause, higher than previously reported (*Kolesnikov-Gauthier et al., 2005*; *Zhang et al., 2009*; *Han et al., 2018*). The positive interpretation of lesions may be the reason for the high accuracy of our research results. In previous studies, only SUVmax and SUVmax combined with serum Ca125 were used as parameters for positive interpretation. In addition to using SUVmax as a positive reference, our study also included anatomical abnormalities only on CT as possible positive lesions.

Due to only two patients having metastasized to the pericardium, no further subgroup analysis was conducted. Previous studies have reported that the sensitivity of $^{18}$F-FDG PET/CT in diagnosing pleural and peritoneal metastasis was 84.4% and 86.4%, respectively

(*Zhang et al., 2009*; *Du, Zhu & Yu, 2022*), similar to our current study's sensitivity of 84.3%. These results are higher than those of *Suzuki et al. (2004)* and *Turlakow et al. (2003)* with sensitivity ranging from 66.7% to 78%. The specificity and accuracy of PET/CT in our study were 93.0% and 88.3%, respectively, both higher than in previous studies, with specificity and accuracy of 80% and 84.6% for pleural metastasis and 83.3% and 85.0% for peritoneal metastasis, respectively (*Zhang et al., 2009*; *Du, Zhu & Yu, 2022*). The possible reason is that in our study, we combined patient clinical data and tumor marker examination results to improve the accuracy of diagnosis.

The current study results demonstrate that PET/CT has high diagnostic performance in detecting primary lesions of unknown serous cavity effusion and detecting serous cavity metastasis, despite the presence of false-negative and false-positive lesions. For primary lesions, there are 10 false-negative and seven false-positive patients. False-negative results occur in mesothelioma, lung adenocarcinoma, fallopian tube adenocarcinoma, hepatocellular carcinoma, MALT lymphoma, and gallbladder signet ring cell carcinoma. Possible causes include low expression of glucose transporter 1 in these pathological types of tumors, resulting in lower uptake of $^{18}$F-FDG on PET/CT (*Soyluoglu & Ozdemir Gunay, 2023*). Moreover, the high glucose-6-phosphatase activity in the tissues of highly differentiated hepatocellular carcinoma can dephosphorylate the phosphorylated $^{18}$F-FDG, resulting in low $^{18}$F-FDG uptake in PET/CT, leading to false negatives (*Sun et al., 2009*). Compared to true positive tumors, the smaller tumor volume of these primary tumors is another possible reason for false negativity. False-positive cases occur in ovarian theca fibroma, organizing pneumonia, tuberculous serositis, and Castleman's disease. From these cases, it can be seen that inflammation and granuloma are the main sources of false positives in our study, and macrophages and neutrophils in inflammatory or granulomatous lesions are metabolically active, resulting in excessive production of glycolytic enzymes that are presented as increased uptake of $^{18}$F-FDG on PET/CT (*Murakami et al., 2013*). Regarding serous cavity metastasis, false negatives are derived from ovarian adenocarcinoma, rectal adenocarcinoma, and lung adenocarcinoma, which may also be caused by the small size of the lesion resulting in low $^{18}$F-FDG uptake on PET, and the serous membrane thickening is not obvious on CT. In three cases of false negative serous cavity metastasis, bacterial infection occurred in the serous cavity, resulting in strong uptake of $^{18}$F-FDG on PET, leading to erroneous judgment.

As one of the commonly used PET/CT parameters, SUVmax serves as a semi-quantitative indicator of glucose consumption, aiding in distinguishing benign and malignant diseases in most cases. ROC curve analysis revealed that the area under the curve of SUVmax is the largest, significantly surpassing those under the curves of Ca125, Ca199, and CEA. This indicates that PET/CT is the optimal method for differentiating malignant from benign serous cavity effusion. However, the distribution figure of SUVmax indicates that elevated SUVmax is also present in some benign diseases, particularly in non-specific infections and tuberculosis. Similarly, certain malignant tumors, such as the false negative pathological types found in our study (*e.g.*, signet ring cell carcinoma, well-differentiated liver cancer, and MALT lymphoma), exhibit lower SUVmax values, overlapping with benign lesions. Therefore, when using the parameter SUVmax of PET/CT to differentiate benign and

malignant serous cavity effusion, it is necessary to combine the patient's clinical history, serum tumor markers, and anatomical changes on CT to enhance diagnostic accuracy.

A limitation of the current research is that, when evaluating serous cavity metastasis using PET/CT, due to the limited number of patients with pericardial metastasis, subgroup analysis by pericardium, pleura, and peritoneum was not conducted. Additionally, not conducting pathological biopsy of all lesions when evaluating serous cavity metastasis is another limitation, and some lesions were identified through follow-up results of CT, MRI, or PET/CT. However, conducting pathological biopsies on all lesions is not ethical and is not feasible.

# CONCLUSION

$^{18}$F-FDG PET/CT proves to be an effective imaging modality for detecting undetermined pleural effusion, including pericardial effusion, pleural effusion and ascites. It demonstrates high diagnostic performance in differentiating primary disease from benign to malignant and evaluating serous cavity metastasis.

**Abbreviations**

| | |
|---|---|
| **CT** | computed tomography |
| **$^{18}$F-FDG** | fluoro-18 fluorodeoxyglucose |
| **PET** | positron emission tomography |
| **MRI** | magnetic resonance imaging |
| **SUVmax** | maximum standardized uptake value |
| **ROI** | region of interest |

## Funding

This study was supported by the Guizhou Province Science and Technology Plan Project (grant numbers: Qiankehe-ZK[2024]-329) and Zunyi Science and Technology Joint Fund (grant number: HZ-2023-284). The funders had no role in study design, data collection and analysis, decision to publish, or preparation of the manuscript.

## Grant Disclosures

The following grant information was disclosed by the authors:
The Guizhou Province Science and Technology Plan Project: Qiankehe-ZK[2024]-329.
Zunyi Science and Technology Joint Fund: HZ-2023-284.

## Competing Interests

The authors declare there are no competing interests.

## Author Contributions

- Xianwen Hu conceived and designed the experiments, performed the experiments, analyzed the data, prepared figures and/or tables, authored or reviewed drafts of the article, and approved the final draft.

- Ya Li performed the experiments, prepared figures and/or tables, authored or reviewed drafts of the article, and approved the final draft.
- Jiong Cai analyzed the data, authored or reviewed drafts of the article, and approved the final draft.
- Pan Wang analyzed the data, prepared figures and/or tables, authored or reviewed drafts of the article, and approved the final draft.

## Human Ethics

The following information was supplied relating to ethical approvals (i.e., approving body and any reference numbers):

Ethics Committee of Affiliated Hospital of Zunyi Medical University.

## Data Availability

The raw data is available in the Supplemental Files.

## Supplemental Information

Supplemental information for this article can be found online at http://dx.doi.org/10.7717/peerj.19495#supplemental-information.

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
