# Peer review of "The role of 18F-FDG PET/CT in patients with serous cavity effusion of undetermined origin: a retrospective clinical study"

_PeerJ, doi:10.7717/peerj.19495_

## Round 0.1 · original submission · Major Revisions

Reviewers 1 and 2 have several important comments. Please respond to them in detail.

Reviewer 1 ·

Basic reporting

Overlapping involvement: Whole body F18FDG-PET scans are the typical imaging modality obtained in patients with suspected metastatic malignancy. As such, all potential sites of involvement (pericardial, pleural and abdomen) are imaged and there is bound to be overlapping involvement of multiple sites. This is also evident in the number of effusions reported and tally of results which exceed the 134 patients in the cohort. It seem that there were a total of 94 malignant tumors, but the reporting throughout the manuscript is jumbled and makes it difficult to determine the exact number of cases that are used to report their detection rates. As an example, in the first part of the results, they report 94 malignant tumor lesions with F18FDG-PET scans detecting 91 lesions and a detection rate of 92.9%. It is unknown what was used for the numerator and denominator as none of the numbers reported would generate the 92.9% detection rate reported.

Reporting of results: Given the confusion generated by their reporting, it would be better if they could report the fraction of their 134 patient cohort with fluid in question (pericardial, pleural and ascites) and then the number of patients with uptake in any of those three sites. The detection percentages reported for all three sites are very similar and it would be more helpful to the reader to know the number of patients with uptake at any site and then the breakdown by site, separated by malignant and non-malignant status.

Experimental design

Population: Their results focus on a highly selected group of patients, and no information is given on the original cohort from which these patients are derived. Were these all consecutive patients with undiagnosed effusions or only who underwent F18FDG-PET scans? Was the diagnosis already known in some patients, and if so, were the F18FDG-PET scans obtained to determine the extent of metastatic disease? It the diagnosis is already known and given the high probability of malignancy, the F18FDG-PET scans would be skewed towards detection which may inflate the accuracy of testing.

Validity of the findings

See above comments. The results are overlapping and it is difficult to discern the actual utility of F18FDG-PET scans with the data they have presented. Changes are suggested above.

Additional comments

GENERAL COMMENTS: The authors present a single center experience with F18FDG-PET scans in the evaluation of unexplained serous cavity effusions. Although they try and separate their analysis into the location of the effusion specifically pericardial, pleural and ascites, their presentation is somewhat muddled and makes it difficult to accurately determine what may be the role of F18FDG-PET scans in these patients. Additional information is needed before making a determination on the utility of their results.

SUV uptake: it is well known that malignant lesions have higher SUV uptake than benign lesions, although there is overlap with active inflammatory conditions such as tuberculosis or organizing pneumonia where the uptake can be comparable to malignant uptake. However very high SUV values are uniformly malignant and their analysis may identify a threshold with overwhelming suggest malignancy. Perhaps they can report a likelihood ratio for malignancy for a high SUV (? 15.0). This point is well established by the ROC curve reported.

More detail is provided in other comment sections.

Annotated reviews are not available for download in order to protect the identity of reviewers who chose to remain anonymous.

·

Basic reporting

The English is clear but can be improved. No other comment.

Experimental design

As a retrospective study, the new contribution of this study over prior studies should be more prominent.

Validity of the findings

For those diagnosed by follow up, should be more clearly stated. How many patients, how can they be diagnosed by follow up, and what's the diagnoses.

Additional comments

1. Title: "The role of 18F-FDG PET/CT in the serous cavity effusion of undetermined origin" should be more clear if changed to "The role of 18F-FDG PET/CT in patients with serous cavity effusion of
undetermined origin"
2. Enroll: How many patients were diagnosed by follow up, how can they be diagnosed by follow up, and what's the diagnoses.
3. Diagnosis: Not clear for the positive diagnosis of 18F-FDG PET/CT. Why "if anatomical and morphological abnormalities were observed in the organ tissue on CT, even in the absence of increased 18F-FDG uptake, it was considered a suspicious positive lesion".
4. More attention was paid to false negatives and false positives in the article, but no typical examples of true positives and true negatives were demonstrated.
5. It's unfair to compare SUV with CA125, CA199, and CEA each alone in a group of patients with different kinds of tumors. Can it compare to the combination of the tumor biomarkers?
6. References: Most of the references are too old. Add rencent publications and delet the old references if not necessary.

·

Basic reporting

No Comment

Experimental design

No comment

Validity of the findings

No comment

Additional comments

The study is well designed and well written with organized and systematized results and clear table and figures with relevant results, detailed discussion and conclusion fulfill the aim of the work.

---

## Round 0.2 · Minor Revisions

The reviewer has some additional comments on your manuscript.

Reviewer 1 ·

Basic reporting

The authors have provided some clarity in their manuscript, but there remains some inconsistencies which need to be corrected to get a full appreciation of their results. In addition, the analysis of their data remains difficult to follow.

In the revision they report a total of 134 patients, of which 21 have pericardial effusions, 94 have pleural effusions and 54 have ascites. Table 1 has slightly different numbers in these categories despite the notation that there are overlapping numbers depending on the type of effusion.

The longstanding issue with FDG-PET scan has been the overlap between malignant and inflammatory conditions. This distinction may not be apparent until a biopsy is performed. Since this is a retrospective review, the assumption is that the authors are aware of the final diagnosis and those with malignancy. Based on their report, this represents 101 tumor lesions, with 94 malignant tumors in 134 patient as listed in table 1. This is different from the 91 malignant lesions they report in their manuscript. This makes it confusing to know exactly how they have based their calculations of sensitivity, specificity, positive and negative predictive value. The report is somewhat difficult to follow and needs to be revised. To start, they should have indicated that of 134 FDG-PET scans for suspected malignancy, x number were abnormal. Of the x number of abnormal scans, 101 were eventually diagnosed with tumors. In other words, how many of the 33 with non-tumor lesions also had abnormal FDG-PET scans, thus confounding the results. Of those with abnormal scans, 94 were eventually determined to have malignant tumors. There should have been the basis (number of abnormal scans) from which they should have calculated their sensitivity, specificity, etc.

After the above analysis, they can also calculate a sensitivity/specificity with an analysis based on the number with malignancy. For example, of the 92 or 94 pleural effusions, 64 were diagnosed with malignancy. Of those 64, how many had an abnormal FDG-PET scan? They should perform the same analysis for pericardial effusion and ascites. It is likely that each body cavity may have different thresholds of detection.

Experimental design

The authors never addressed previous questions about their experimental design.

A prospective analysis provides a true experience reflects the true impact of the PET scan. It is unclear if the represented consecutive patients or selected patients. Their analysis is subject to gaps in their data unless it involves ALL of their patients who underwent a PET scan with suspected malignancy. This is not evident in their methods.

Were some of the scans obtained for patients who were NOT suspected to have an underlying malignancy? Or were all of the patients scanned because of suspected malignancy? Sensitivity and specificity are greatly influenced by the cohort of patients undergoing investigation, so these are important areas to disclose.

Validity of the findings

It is difficult to assess the validity of their findings based on their presentation and discrepancy with the numbers used in their calculations. PET scans are effective in the assessment of serous cavity effusions, but it is not clear if the ability to detect malignancy is as high as they have reported.

Additional comments

No additional comments.

---

## Round 0.3 · accepted · Accept

Two referees have examined the revised manuscript and agree that the concerns raised in the previous review have been satisfactorily addressed.

Reviewer 1 ·

Basic reporting

Satisfactory

Experimental design

Satisfactory

Validity of the findings

Satisfactory presentation

Additional comments

This study is limited by its retrospective study design, and the sensitivity, specificity, positive and negative predictive values are tempered by the fact that their study only included those whose diagnosis could not be established by any other means. This means that the full pool of potential subjects with suspected malignancy who might undergo FDG-PET imaging is unknown. In other words, this is a selected group of patients, and their experience would be difficult to generalize to other populations since the evaluation for these patients would vary from institution to institution.

·

Basic reporting

No new comment or suggestion.

Experimental design

No new comment or suggestion.

Validity of the findings

No new comment or suggestion.

Additional comments

No new comment or suggestion.